# Efficacy of Continuous Lumbar Plexus Blockade in Managing Post-Operative Pain after Hip or Femur Orthopedic Surgeries: A Systematic Review and Meta-Analysis

**DOI:** 10.3390/jcm13113194

**Published:** 2024-05-29

**Authors:** Wijdan A. AlMutiri, Ebtesam AlMajed, Muath M. Alneghaimshi, Afnan AlAwadh, Reem AlSarhan, Malak N. AlShebel, Rayan Abdullah M. AlMatrody, Rafa Hadaddi, Reem AlTamimi, Rawan Bin Salamah, Lama A. AlZelfawi, Saud K. AlBatati, Alanood AlHarthi, Ghayda AlMazroa, Abdullah M. AlHossan

**Affiliations:** 1College of Medicine, Princess Nourah bint Abdulrahaman University, Riyadh 11671, Saudi Arabia; 439003552@pnu.edu.sa (E.A.); 439001114@pnu.edu.sa (A.A.); 439001996@pnu.edu.sa (R.A.); 439005046@pnu.edu.sa (R.A.); 439000602@pnu.edu.sa (R.B.S.); 439001449@pnu.edu.sa (L.A.A.); 439000212@pnu.edu.sa (A.A.); 2Orthopaedic Surgery Department, King Fahad Military Medical Complex, Dhahran 31932, Saudi Arabia; muathalneghaimshi@gmail.com (M.M.A.); saudalbatati@hotmail.com (S.K.A.); alhossanmd@gmail.com (A.M.A.); 3College of Medicine, King Saud Bin Abdulaziz University for Health Sciences, Riyadh 11481, Saudi Arabia; alshebel249@ksau-hs.edu.sa; 4Neurosurgery Department, King Fahad University Hospital, Alkhobar 34445, Saudi Arabia; ralmatrody@hotmail.com; 5College of Medicine, Jazan University, Jazan 45142, Saudi Arabia; 201700042@stu.jazanu.edu.sa; 6College of Medicine, Qassim University, Qassim 51452, Saudi Arabia; 352206485@qu.edu.sa

**Keywords:** lumbar plexus block, orthopedic surgery, pain management

## Abstract

**Background:** Post-operative pain management is essential for optimizing recovery, patient comfort, and satisfaction. Peripheral nerve blockade, or lumbar plexus block (LPB), has been widely used for analgesia and regional anesthesia. This study explored the existing literature to determine the efficacy of continuous lumbar plexus blockade in managing post-operative pain following hip or femur surgery. **Methods:** Reviewers comprehensively searched electronic databases to identify peer-reviewed scholarly articles reporting the efficacy of lumbar plexus block in managing post-operative pain after orthopedic surgery. The potential articles were carefully selected and assessed for the risk of bias using the Cochrane Collaboration Risk of Bias assessment tool. Data were systematically extracted and analyzed. **Results:** The literature search yielded 206 articles, 20 of which were randomized controlled trials. Lumbar plexus block demonstrated superior pain relief compared to conventional pain management approaches like general anesthetics. In addition, LPB reduced patients’ overall opioid consumption compared to controls, reduced adverse effects, and enhanced functional recovery, which underlines the broader positive impact of meticulous pain management. More patients could walk more than 40 feet after the second day post-operatively among the lumbar plexus group (14.7%) compared to the continuous femoral group (1.3%). Other parameters, including cortisol levels and hemodynamic stability, were evaluated, showing comparable outcomes. **Conclusions:** Lumbar plexus block is effective in pain management after orthopedic surgery, as shown by the lower pain scores and less opioid consumption. Additionally, patient satisfaction was relatively higher in LPB-treated patients compared to other approaches like general anesthesia.

## 1. Introduction

Post-operative pain management continues to pose a significant challenge to the medical community [1]. The benefits of using traditional opioid pain-relieving methods lie in the quick pain relief and ease of administration through different routes, making them suitable for various situations. However, opioids may induce various side effects that include nausea, vomiting, constipation, dizziness, sedation, and respiratory depression, which may affect patient recovery and satisfaction. Addiction and dependence on opioids are also a concern. Therefore, an exploration of alternative approaches to pain management may be of benefit [2]. Over the past several decades, peripheral nerve blockade, or lumbar plexus block (LPB), has been widely used for analgesia and regional anesthesia [3]. However, its routine use has been limited due to a lack of experience and unidentifiable risks [4]. Initially introduced in 1973, the LPB affects the main branches of the hip joint capsule nerves, including the femoral, lateral femoral cutaneous, and obturator nerves [5].

A study conducted at a large tertiary, university-affiliated hospital aimed to shed light on the differences in post-operative pain between patients undergoing orthopedic surgery or laparotomy under general anesthesia over one year, in which they compared the intensity of immediate pain experienced by both groups, as well as the prevalence and effectiveness of the routine analgesic protocol of morphine, ketamine, and diclofenac administration [6]. Interestingly, the study revealed that orthopedic patients, on average, endured more intense pain immediately following surgery and subsequently required higher doses of analgesic medications compared to patients who underwent laparotomy procedures. Accordingly, tailored post-operative pain management is necessary to optimize patient comfort and recovery.

Post-operative pain is common in hip and femur orthopedic surgeries, impacting patient recovery and potentially causing complications. Post-operative pain is a common complication in hip and femur orthopedic surgeries, and its impact can be significant on patient recovery. Elderly patients are particularly susceptible to this pain, which can delay their recovery and increase the risk of complications. According to a study published in PubMed, post-operative pain was found to be a significant predictor of prolonged hospital stays among elderly patients following hip and femur surgeries. The findings of this study emphasize the importance of effective pain management strategies in elderly orthopedic patients to optimize their recovery and prevent potential complications [7]. The results of another article published in PubMed on elderly patients with hip fractures showed a lesser decline in walking ability and better functional recovery, which probably led them to remain in their own homes alone without assistance. In addition, the orthogeriatric assessment explained 8% of the variability of ADL score (using multivariate regression models) and emerged as independent predictors of 6-month walking ability on multivariate logistic analysis. Belonging to the orthogeriatric group led to an almost 4-fold higher probability of autonomous ambulation 6 months later [8]. 

Therefore, effective pain management is crucial, and regional anesthesia techniques like continuous lumbar plexus block (CLPB) have emerged as promising alternatives to traditional opioid-based regimens. Amiri et al. conducted a study to evaluate the safety, efficacy, and acceptability of LPB in the anesthesia management of elderly patients undergoing open reduction and internal fixation (ORIF) of hip fractures [9]. The study sample consisted of 50 patients in a tertiary educational hospital. In this study, LPB proceeded with light intravenous sedation using midazolam 0.15–0.3 mg/kg found to be an effective, safe, and acceptable anesthetic option for managing hip fractures in older people.

Surgical patients with suboptimal acute post-operative pain management suffer a variety of negative consequences. They are associated with increased morbidity, functional impairment, delayed recovery time, prolonged use of opioids, and higher healthcare costs. These consequences can have a lasting impact on a patient’s overall recovery and well-being. Therefore, pain management is essential to surgical procedures during and after surgery [10,11,12,13].

Despite the increasing adoption of CLPB, the evidence base remains fragmented and inconclusive. A systematic review will synthesize and critically evaluate research from various studies, providing a comprehensive overview of the current evidence and identifying areas of uncertainty or conflicting findings. This will facilitate the optimization of clinical practice guidelines and guide future research. In this systematic review and meta-analysis, we evaluated the existing literature to determine the effectiveness of continuous lumbar plexus blockade in managing post-operative pain following hip or femur surgery.

## 2. Materials and Methods

This study was carried out according to the Preferred Reporting Items for Systematic Review and Meta-analysis (PRISMA) guidelines [14]. It is registered in the International Prospective Register of Systematic Reviews (PROSPERO) (ID: CRD42023492890).

### 2.1. Eligibility Criteria

This study included RCTs in the English language with participants aged 18 years or older, assessing lumbar plexus blockade as a method of post-operative pain management in patients undergoing hip or femur orthopedic surgeries, such as hip or knee replacements, arthroscopic hip surgeries, surgery for femoral fractures. Additionally, studies whose primary outcomes measured were pain scores and the need for additional analgesia methods were included without time restrictions. On the other hand, irrelevant study designs, non-English language studies, and studies that did not directly involve orthopedic hip or femur patients were excluded.

### 2.2. Search and Identification of Studies

A comprehensive search was conducted across the following databases: Google Scholar, ProQuest, EMBASE, PubMed, Web of Science, and Emerald. Only relevant English randomized controlled trials (RCTs) were included in this review. Our search terms were as follows: (“Lumbosacral Plexus” OR “Lumbar plexus” OR “Posterior Femoral Cutaneous Nerve” OR “Inferior Cluneal Nerves” OR “Lumbar Plexus Blockade” OR “Nerve Blockade” OR “NerveBlock” OR “Anesthesia”) AND (“Post-operative Pain” OR “Post-surgical Pain” OR “Pain Management” OR “Pain Control” OR “Patient Satisfaction” OR “Analgesics” OR “Visual Analog Scale” OR “Postoperative Analgesia”)AND (“Hip” OR “Hip Fractures” “Fracture Dislocation of the Hip” OR “Hip Trauma” OR “Hip arthroplasty” OR “Hip arthroscopy” OR “Hip replacement” OR “Hip Prosthesis” OR ”Femur” OR “Trochanter” OR “Femur Fractures” OR “Intertrochanteric Fractures” OR “Trochlear Fractures” OR “Trochanteric Fractures” OR “Subtrochanteric Fractures”).

### 2.3. Study Selection

After the primary search, the yielded studies were imported to the Rayyan platform for de-duplication and initial screening by title and abstract, which was done independently by two authors (R.S, E.A). A third party resolved any disagreement. The included articles were retrieved and screened in full text by three independent reviewers who independently evaluated full-text articles that met inclusion and exclusion criteria (A.A, M.S, E.M). For an article to be considered for inclusion, the article needed to be approved by the independent reviewers.

### 2.4. Data Extraction

The data for this study were independently extracted by four authors (L.Z, R.B, G.M, A.H). Data that were extracted from the studies included study characteristics, participant details, intervention details, comparison/control group details, outcomes measured, key findings, adverse events if reported, and study limitations. The extracted data were systematically tabulated in a predesigned Excel workbook using Microsoft Excel 2021.

### 2.5. Methodological Quality Assessment

The Cochrane risk tool for bias was used to evaluate the quality of RCTs. The tool encompasses various domains, with each domain’s judgments contributing to an overall RoB2 judgment that spans five main domains. These domains are fixed, focusing on aspects of trial design, conduct, and reporting using a series of “signaling questions” to elicit information relevant to the risk of bias. This is then judged using an algorithm, and the judgments can be “low” (for all domains, the risk of bias is low), “some concerns” (for at least one of the domains, there is some concern), or “high” (for at least one domain, there is a high risk or some concerns for multiple domains). Two authors (R.B. and S.B.) independently conducted the risk of bias assessment, and after consulting with senior authors (M.N. and A.H.), they resolved disagreements through consensus [15].

### 2.6. Data Analysis

Extracted data were procedurally analyzed according to the predominant themes [16]. In addition, the quantitative data were exported to Review Manager Version 5.4.1 and analyzed using an intervention review approach. Moreover, meta-analyses were conducted using the random effects analysis approach, the Mantel–Haenszel statistical method, and the odds ratio as the effect measure. A 95% confidence interval was applied in the analyses. We assessed study heterogeneity using the inconsistency index [13], which corresponded to 0–30%, 31–60%, 61–75%, and 76–100% representing low, moderate, substantial, and considerable heterogeneity. In order to assess both primary and secondary outcomes, corresponding forest plots were constructed. Subgroup analyses were performed for the following when data were available: VAS pain scores between LPB-treated patients and controls, NRS pain scores between LPB-treated patients and controls, cortisol levels in LPB-treated patients and control, incidences of nausea among LPB patients, systolic blood pressure in LPB-treated patients and controls, diastolic blood pressure in LPB-treated patients and controls, and heart rate in LPB-treated patients and controls.

## 3. Results

### 3.1. Study Selection

The literature search yielded 206 articles, of which 20 remained after applying the inclusion criteria. The study selection process is detailed in the PRISMA flow chart in Figure 1. 

The included randomized controlled trials were high-quality (Figure 2 and Figure 3) and primarily focused on patients undergoing hip surgeries, including hip arthroplasty, hip arthroscopic surgery, and proximal femoral fractures. The controls employed several techniques, including fascia iliaca block (FIB), femoral nerve block (FNB), quadratus lumborum block (QLB), Supra-Inguinal Fascia Iliaca Block (SIFIB), Paravertebral Block (PVB), Lumbosacral Plexus Block (LSPB), Lumbar Erector Spinae Plane Block (LESPB), PCA (Patient-Controlled Analgesia), and PCEA (Patient-Controlled Epidural Analgesia). The sample sizes ranged from 34 to 225 with a total of 1687 patients, and the studies were conducted in seven countries, with the majority being in the USA; ten studies [16,17,18,19,20,21,22,23,24,25] were conducted in the USA, two [26,27] in Switzerland, two [28,29] in India, two [30,31] in China, two [32,33] in Serbia, one [34] in Chile, and one [15] in Egypt. The trials utilized pain scores, including VAS pain scores, NRS pain scores, and opioid consumption, as crucial outcome measures to examine the efficacy of analgesic interventions. The trials were conducted between 2003 and 2021, and durations spanned 6–30 months. The details of the included studies are presented in Table 1 below.

### 3.2. Pain Scores 

In their respective studies, Anis et al. [34], Badiola et al. [17], and Bravo et al. [18] assessed the efficacy of various analgesic techniques in post-operative pain management. Anis et al. [34] observed significantly higher Visual Analog Scale (VAS) pain scores in subjects without LPB from the first hour post-operatively compared to those treated with the LPB. Badiola et al. compared FIB with LPB and reported insignificant differences in pain scores 15 min post-operatively [17].

In addition, Goytizolo et al. reported significantly lower pain scores in the LPB group compared to the control group on the second day post-operatively, with medians of 1.5 and 2.5, respectively [19].

Ilfeld et al. examined the differences between femoral and lumbar plexus groups, revealing statistically insignificant variations in pain scores (3.5 ± 1.8 for lumbar plexus and 3.6 ± 1.8 for femoral). The study further investigated continuous posterior LPB versus periarticular injection with ropivacaine, finding no statistically significant differences in pain scores [21].

Pain scores were comparable among the two groups in Johnson et al. [32]. According to Kendrisic et al., central nerve block patients exhibited the lowest average pain scores. At the same time, LPB members reported significantly lower pain scores than the femoral group after one and two days post-operatively [33].

ESPB emerges as a noteworthy analgesic technique, demonstrating a more significant reduction in pain scores than LPB. Specifically, ESPB reduced by 1.413 ± 0.795 compared to 0.960 ± 0.663 observed in the LPB group [29]. LPB significantly reduced the pain scores until six hours after randomization (VAS 1.4 ± 1.3 against the control group 2.4 ± 1.4) [27].

As shown in Figure 4, LPB-treated patients had significantly lower pain scores (VAS) than controls without LPB, with a *p*-value of 0.02. Conversely, pain scores (NRS) exhibited high heterogeneity, as seen in Figure 5. The differences in pain scores among LPB-treated patients and controls without LPB were statistically insignificant, with a *p*-value of 0.39.

### 3.3. Opioid Use

Interestingly, the researchers noted less opioid use among LPB patients compared to FIB, although the difference was statistically insignificant [18,20,34]. Bravo et al. (2020) explored median cumulative morphine use, revealing no significant difference between LPB and SIFIB groups [18].

Moreover, in Goytizolo et al., opioid consumption exhibited comparability between the experimental and control groups [19]. Contrarily, QLB and LPB patients showed similar opioid use during the first 24 h post-operatively [20,22].

According to Siddiqui et al. [25] and Stevens et al. [27], the LPB group and plexus group, respectively, required less morphine than the patient-controlled analgesia. On the other hand, Ilfeld et al. [21] and Johnson et al. [32] found no statistical differences in opioid use. 

### 3.4. Cortisol Levels

Patients without LPB exhibited high blood cortisol levels after the second post-operative hour, contrasting with those treated with the block who experienced a delayed increase after the sixth post-operative hour [34]. Serum cortisol levels in the PNB group were comparable to those in the general anesthesia–PCA morphine group. Additionally, the cortisol values in the central nerve block group were similar to those from the spinal anesthesia–PCA morphine group after four hours post-operatively. However, after 12 h, the general anesthesia–PCA morphine registered the highest serum cortisol level, while the central nerve block group displayed the lowest serum cortisol level [23].

The overarching cortisol levels after 12 h post-operatively illustrate heterogeneity and an overall statistically insignificant difference in the concentration between the LPB group and the controls (*p*-value 0.35). Figure 6 shows a forest plot comparing cortisol levels among LPB-treated patients and the controls.

### 3.5. Adverse Effects 

In an initial observation, all the study participants reported no side effects during the first 24 h post-operatively, irrespective of the use of LPB. [34] Adverse effects among the lumbar plexus and control groups were similar, with patients reporting itching and nausea [18].

Notably, despite episodes of nausea, urine retention, and hypotension across various treatment groups, including central nerve block (CNB), PNB, general anesthesia–PCA morphine, and spinal anesthesia–PCA morphine, LPB-treated patients demonstrated significantly lower incidences of nausea [23,33]. Diwan et al. reported no hemodynamic instability or adverse respiratory effects [29]. 

According to Siddiqui et al., the patient-controlled analgesia group experienced more nausea and vomiting than the lumbar plexus group, with similar pruritus severity among the groups [25]. There was one reported case of epidural spread in the lumbar plexus group [36]. There were significantly fewer incidences of nausea among LPB-treated patients exhibiting a *p*-value less than 0.00001. Figure 7 shows a proportion meta-analysis of the incidences of nausea among LPB-treated patients.

### 3.6. Patient Satisfaction

Badiola et al. found a comparable quality of recovery among FIB and LPB patients [17]. The femoral group reported a satisfaction score of 8.5 out of 10, while the LPB group reported a 10 out of 10, indicating high satisfaction for both techniques [21]. Comparable patient satisfaction between pericapsular injection and lumbar plexus blockade [24,29]. Conversely, LPB showed significantly higher patient satisfaction compared to Continuous Epidural Analgesia (CEA), highlighting the superiority of LPB in patient-reported outcomes. Higher satisfaction scores were reported among central nerve block (CNB), PNB, general anesthesia–PCA morphine, and spinal anesthesia–PCA morphine, LPB-treated patients [23,33].

### 3.7. Heart Rate Parameters and Blood Pressure

Due to analgesic administration, patients treated without the LPB experienced a standard range of systolic and diastolic blood pressures, heart rate (HR), and respiratory rate (RR). However, the systolic and diastolic blood pressures increased after the sixth hour post-operatively, demonstrating sympathetic stimulation due to post-operative pain corresponding with high VAS scores. Additionally, the blood pressure increased again after the 12th hour [34].

On the other hand, subjects treated with LPB experienced a standard range of systolic and diastolic blood pressures, RR, and HR since they were pain-free. However, the heart rate variables increased after the sixth post-operative hour, corresponding to the pain score [34]. There was an insignificant difference in the systolic blood pressure with a *p*-value of 0.36, as shown in Figure 8 [34,36]. Similarly, the diastolic blood pressure was lower in the LPB-treated group. However, the difference was statistically insignificant, as shown in Figure 9 [34,36]. Moreover, the heart rate was comparable among LPB-treated patients and controls, with a *p*-value of 0.69, as shown in Figure 10.

### 3.8. Quality of Recovery

More patients could walk more than 40 feet after the second day post-operatively among the lumbar plexus group (14.7%) compared to the continuous femoral group (1.3%) [28]. Quadriceps muscle strength was significantly increased in the quadratus lumborum group [36].

### 3.9. Length of Hospitalization

The suprainguinal fascia illiaca block (SIFIB) group was associated with a shorter hospital stay than the LPB group [18]. There was a comparable length of hospital stay between the femoral infusion group and the lumbar plexus group [21].

## 4. Discussion

Post-operative pain management is essential in optimizing the quality of recovery, patient comfort, and satisfaction [37]. Several superior qualities of LPB contribute to the reduction in post-operative pain, including effective pain relief post-operatively, which reduces the use of opioids and other systemic analgesics. The included studies compared LPBB with various pain management approaches, including quadratus lumborum block, suprainguinal fascia iliaca block, general anesthesia–PCA morphine, and spinal anaesthesia–PCA morphine. The results of this review demonstrate consistency in the efficacy of LPB compared to controls treated without LPB. However, some included studies reported insignificant differences in pain alleviation following LPB intervention. The heterogeneity in the reported results may be associated with the differences in study settings and assessment approaches [38].

On the other hand, LPB consistently demonstrated reduced opioid use [28,29,30,34,36]. Different studies reported varied durations before first opioid use, with LPB patients requiring opioid intervention at a relatively delayed rate compared with other conventional analgesic options. The trends in opioid use were consistently lower in LPB-treated patients, corresponding to the lower pain scores among the patients. LPB is a promising approach to optimizing patient recovery by minimizing the adverse events associated with the use of conventional opioids and analgesic approaches, including respiratory depression, nausea, and vomiting. Joint mobility is essential in optimizing the quality of recovery and functional outcomes following orthopedic surgery [39]. LPB-treated patients demonstrated less hospital stay [18]. However, some studies reported comparable lengths of stay with statistically insignificant differences in the time to discharge [21].

Complications are often associated with surgical procedures and the pain management approaches utilized, including opioids [40]. The present study evaluated the effect of using LPB on the incidences of adverse effects. Studies reported various side effects, including respiratory depression, nausea, and itching. The current study demonstrates that LPB is a relatively safe approach for pain management with fewer reported side effects. However, some included studies reported incidences of side effects for LPB that were comparable to other controls. Patient satisfaction outcomes consistently favor the use of LPB, demonstrating its acceptance and potential of LPB in pain management after orthopedic surgery. The high rates of satisfaction are potentially due to adequate pain relief, minimal opioid requirements, and optimal quality of recovery, which enhance patient experience [41].

Evaluating cardiovascular function and stress levels through serum and blood cortisol levels is paramount in understanding the psychological impact of pain management approaches following orthopedic surgery. The observed rise in cortisol levels after the sixth hour post-operatively signifies a delayed stress response in LPB-treated patients, aligning with adequate pain relief in the early post-operative period. LPB is associated with a stable heart rate, indicating successful pain control. Anis et al. [34] and Yuan et al. [36] reported statistically insignificant differences in heart rate parameters, yet the observed trend suggests a potentially practical approach to maintaining hemodynamic stability post-operatively. Evaluating cardiovascular function and stress levels through serum and blood cortisol levels is paramount in understanding the psychological impact of pain management approaches following orthopedic surgery. The observed rise in cortisol levels after the sixth hour post-operatively signifies a delayed stress response in LPB-treated patients, aligning with adequate pain relief in the early post-operative period. LPB is associated with a stable heart rate, indicating successful pain control. Anis et al. [34] and Yuan et al. [36] reported statistically insignificant differences in heart rate parameters, yet the observed trend suggests a potentially practical approach to maintaining hemodynamic stability post-operatively.

## 5. Clinical Implications

LPB offers a practical approach to post-operative pain management in orthopedic surgery. Still, optimizing outcomes demands careful consideration of various factors, including individual patient needs, specific surgical procedures, and patient preferences. The successful implementation of LPB mandates expertise in anesthetic techniques, ensuring a tailored and personalized care approach that aligns with diverse patient requirements. Notably, the significance of effective pain management extends beyond immediate comfort, playing an essential role in improving overall recovery and reducing the length of hospital stays. The correlation between lower pain scores, reduced opioid usage, fewer adverse effects, and enhanced functional recovery underlines the broader positive impact of meticulous pain management.

As more studies are conducted, we recommend that future studies investigating the efficacy of continuous lumbar plexus blockade include a well-defined control group, such as a group receiving no block, a sham block, or a standard pain management regimen. This will allow for a more efficient evaluation of the intervention’s effect on post-operative pain, the length of hospital stay, and patient satisfaction. Further evaluation of the analgesic effectiveness of continuous lumbar plexus blockade after orthopedic surgery is warranted with larger sample sizes and standardized endpoints, along with exploring potential moderators. 

## 6. Limitations

This study has a few limitations. First, the heterogeneity in the reported results may be associated with different factors, including study settings, comorbidities, surgical procedure differences, and other demographic characteristics. Although the Cochrane Risk of Bias tool was used to assess the quality of the included studies, it is also possible that some studies may still have methodological limitations or biases leading to variability. In addition, this review only includes studies published in the English language, excluding studies published in other languages, which may lead to language bias. Therefore, it is essential to recognize that the identified limitations may impact the overall generalizability of the results. 

## 7. Conclusions

This study aimed to evaluate the efficacy of continuous LPB in pain management after orthopedic surgery. It explored pain scores, adverse effects, and patient satisfaction, demonstrating consistently superior results to the other pain management approaches. LPB showed significant advantages, including superior pain relief, reduced opioid consumption, and enhanced patient satisfaction. The lower pain scores and reduced opioid consumption associated with LPB indicate its ability to optimize patient comfort and recovery. Moreover, the relatively higher patient satisfaction observed in LPB-treated patients compared to methods like general anesthesia emphasizes the importance of considering patient preferences and requirements in pain management planning.

The implications of these findings for clinical practice are substantial. LPB can be considered a valuable alternative to traditional opioid-based regimens, potentially reducing the risk of adverse effects associated with opioids. Implementing LPB in orthopedic surgery protocols has the potential to improve patient outcomes, including faster recovery, reduced morbidity, and enhanced functional outcomes.

However, it is crucial to acknowledge that further research is needed to address the remaining gaps in knowledge. While LPB has shown encouraging results, future studies should focus on specific patient populations, optimal techniques and dosages, and long-term outcomes. Furthermore, examining the applicability and cost-effectiveness of utilizing LPB in various healthcare environments will offer insightful information for medical procedures.

## Figures and Tables

**Figure 1 jcm-13-03194-f001:**
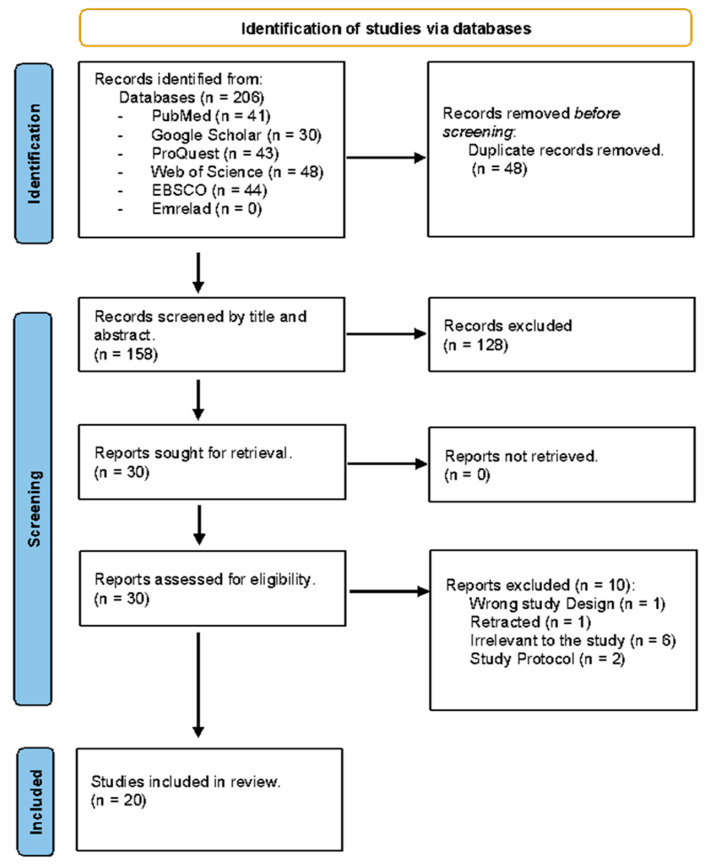
PRISMA flow diagram.

**Figure 2 jcm-13-03194-f002:**
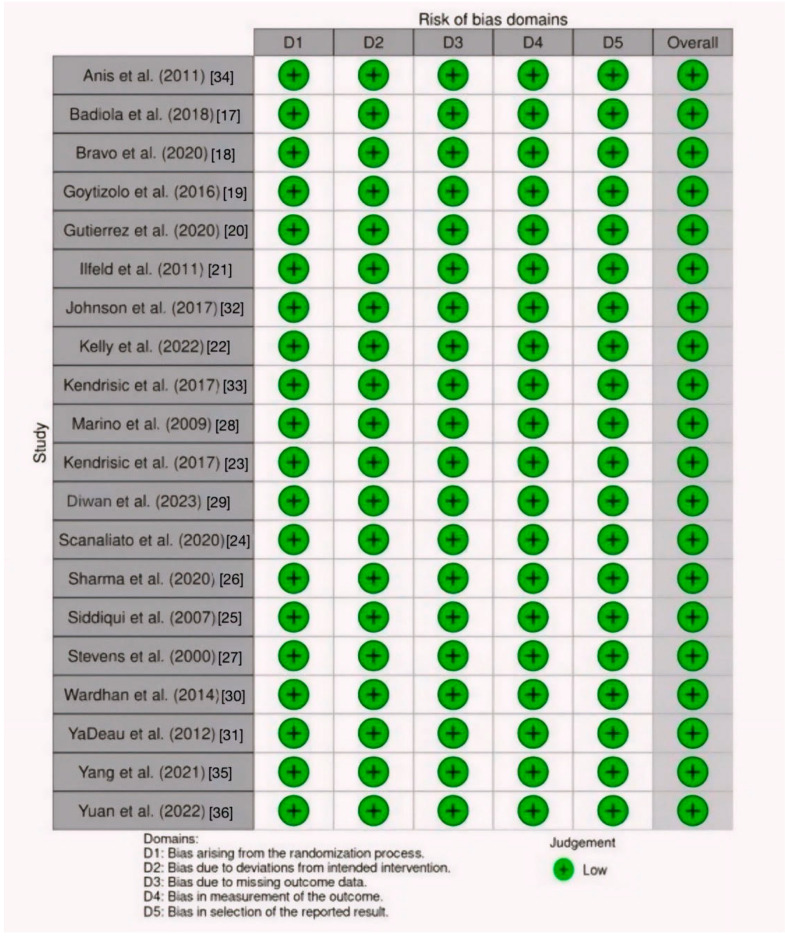
Risk of bias assessment summary.

**Figure 3 jcm-13-03194-f003:**
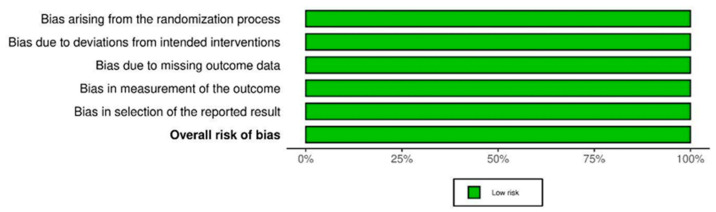
Risk of bias assessment graph.

**Figure 4 jcm-13-03194-f004:**
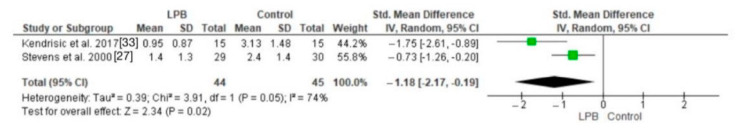
Forest plot comparing VAS pain scores between LPB-treated patients and controls without LPB.

**Figure 5 jcm-13-03194-f005:**
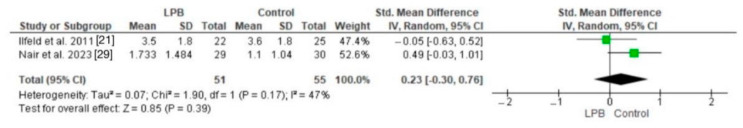
Forest plot comparing NRS pain scores between LPB-treated patients and controls without LPB.

**Figure 6 jcm-13-03194-f006:**
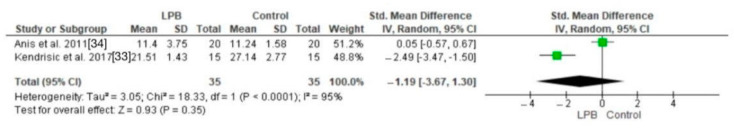
Forest plot comparing cortisol levels in LPB-treated patients and control groups without LPB [33,34].

**Figure 7 jcm-13-03194-f007:**
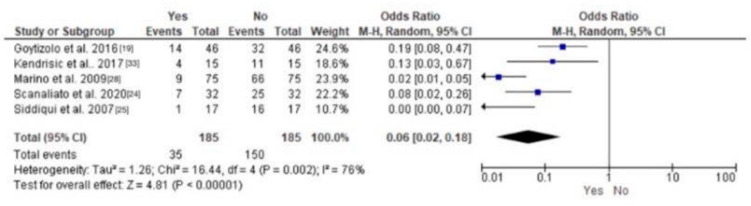
Forest plot showing the incidences of nausea among LPB patients.

**Figure 8 jcm-13-03194-f008:**
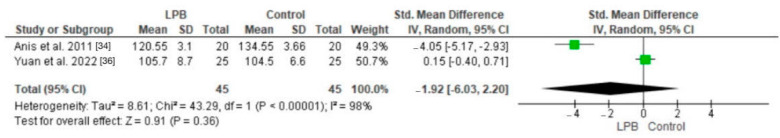
Forest plot comparing systolic blood pressure in LPB-treated patients and controls.

**Figure 9 jcm-13-03194-f009:**
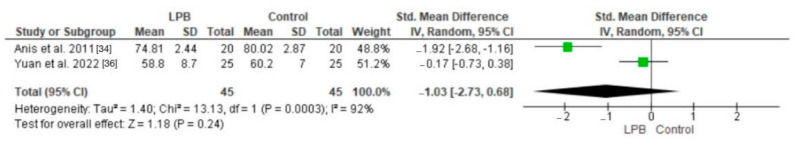
Forest plot comparing diastolic blood pressure in LPB-treated patients and controls.

**Figure 10 jcm-13-03194-f010:**
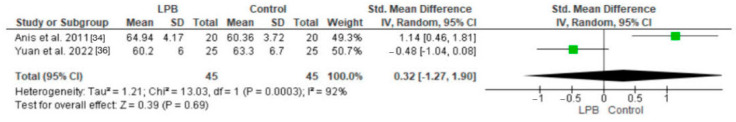
Forest plot comparing heart rate in LPB-treated patients and controls.

**Table 1 jcm-13-03194-t001:** General characteristics of the included studies.

Study	Sample Size	Age	Study Duration	Country	Type of Surgery\Technique	Anesthetic	Outcome Measures
Anis et al. (2011) [34]	60	18–60	Unspecified	Egypt	Hip surgery; posterior LPB	15 mL bupivacaine 0.5% Clonidine 2.5 lg/mL	Pain scores (VAS)
Badiola et al. (2018) [17]	50	>18	13 months	USA	Hip arthroscopic surgery; FIB vs. LPB	30 mL of 0.25% bupivacaine with 1:200,000 epinephrine	Pain scores
Bravo et al. (2020) [18]	60	18–80	Unspecified	Chile	Total hip arthroplasty; SIFIB vs. LPB	Ultrasound-guided lumbar plexus block 40 mL of levobupivacaine 0.25% with epinephrine 5 μg/mL	Pain scores, opioid consumption
Goytizolo et al. (2016) [19]	92	60–100	Unspecified	USA	Total hip arthroplasty; LPB	35 mL of 0.5% Bupivacaine 5 mg of IV midazolam 60 mg of 1.5% mepivacaine 2% lidocaine in 3 mL IV propofol 2–4 mg/kg	Pain scores
Gutierrez et al. (2020) [20]	46	19–90	11 months	USA	Hip arthroplasty; QLB vs. LPB	Hyperbaric 0.75% bupivacaine (10.5–12 mg) Propofol infusion at 50–100 mcg/kg/ min ketamine 20 mg bolus	Pain score, quadriceps strength, and opioid consumption
Ilfeld et al. (2011) [21]	50	≥18	10 months	USA	Total hip arthroplasty; Continuous FNB vs. continuous LPB	Perineural ropivacaine, 0.2% (basal 6 mL/h, bolus 4 mL, 30 min lockout)	Pain scores
Johnson et al. (2017) [32]	159	>18	15 months	USA	Total hip arthroplasty; PNB	Bupivacaine 0.5% with 1:200,000 epinephrine 30 mL bolus Infusion of bupivacaine 0.2%	Pain scores
Kelly et al. (2022) [22]	103	>18	11 months	USA	Total hip arthroplasty; PNB	Ropivacaine (20 mL, 0.5%)	Opioid consumption
Kendrisic et al. (2017) [33]	60	59 ± 9	Unspecified	Serbia	Total hip arthroplasty; LPB vs. epidural analgesia	20 mL levobupivacaine 0.25%	Pain scores (VAS) at rest and on moving
Marino et al. (2009) [28]	225	18–80	30 months	USA	Total hip arthroplasty; Continous LPB vs. FIB	Bolus of 0.6 mL/kg of 0.5% ropivacaine	VAS pain scores
Kendrisic et al. (2017) [23]	60	Unspecified	Unspecified	Serbia	Surgical stress response following hip arthroplasty regarding choice of anesthesia and post-operative analgesia	20 mL 0.25% levobupivacaine	Serum cortisol, insulin, glucose, CRP, incidence of hypotension
Diwan et al. (2023) [29]	70	≥65	12 months	India	Proximal femoral nail for proximal femoral fractures; LESPB vs. LPB	0.2% ropivacaine and 30 mcg clonidine	Pain scores
Scanaliato et al. (2020) [24]	64	17–49	6 months	USA	Hip arthroscopy; Pericapsular injections vs. LPB	40 mL 0.375% ropivacaine	NRS pain scores
Sharma et al. (2020) [26]	50	20–80	Unspecified	India	Total hip replacement; LPB vs. EA	15 mL 0.5% ropivacaine	NRS pain scores
Siddiqui et al. (2007) [25]	34	18–80	Unspecified	USA	Hip arthroscopy; LPB vs. systemic opioids	20 cc 0.25% bupivacaine	Perioperative opioid requirement
Stevens et al. (2000) [27]	60	66 ± 10	Unspecified	Switzerland	Total hip arthroplasty; LPB	0.4 mg/kg 0.5% bupivacaine	Pain reduction
Wardhan et al. (2014) [30]	60	18–75	29 months	USA	Total hip arthroplasty; LPB vs. L2 PVB	15 mL 0.5% ropivacaine	Post-operative opioid consumption and post-operative pain scores
YaDeau et al. (2012) [31]	60	35	Unspecified	Switzerland	Total hip arthroplasty; LPB	0.4 mL/kg 0.5% bupivacaine	Pain scores at rest and morphine consumption
Yang et al. (2021) [35]	167	40–80	10 months	China	Elective total hip arthroplasty; LSPB	0.5% ropivacaine	QoL scores, Mobility, Self-Care, Usual Activities, Pain/Discomfort, and Anxiety/Depression.
Yuan et al. (2022) [36]	50	18–60	6 months	China	Hip arthroscopic surgery; QLB vs. LPB	0.4 mL/kg ropivacaine	VAS pain scores

All studies adopted a randomized controlled trial (RCT) design. LPB (Lumbar Plexus Block), PNB (Peripheral Nerve Block), FIB (Fascia Iliaca Block), SIFIB (Supra-Inguinal Fascia Iliaca Block), QLB (Quadratus Lumborum Block), PVB (Paravertebral Block), LSPB (Lumbosacral Plexus Block), LESPB (Lumbar Erector Spinae Plane Block), VAS (Visual Analog Scale), NRS (Numeric Rating Scale), EA (Epidural Analgesia), CRP (C—reactive protein), USA (United States of America).

## Data Availability

No datasets were generated or analyzed during the current study.

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
