# Peer review of "Efficacy of Continuous Lumbar Plexus Blockade in Managing Post-Operative Pain after Hip or Femur Orthopedic Surgeries: A Systematic Review and Meta-Analysis"

_jcm, 2024, doi:10.3390/jcm13113194_

Round 1

Reviewer 1 Report

Comments and Suggestions for Authors

Title: Efficacy of Continuous Lumbar Plexus Blockade in Managing Post-operative Pain After Hip or Femur Orthopedic Surgeries: A Systematic Review and Meta-Analysis

Abstract:

The abstract presents a clear overview of the study's objectives, methods, key findings, and conclusions. It effectively summarizes the efficacy of continuous lumbar plexus blockade (CLPB) in managing post-operative pain following hip or femur surgery. To improve it further, you might consider incorporating specific findings or metrics from the results section to provide a more comprehensive overview of the study's findings. Additionally, emphasizing the clinical implications of the study's findings could enhance the abstract's impact.

Introduction:

The introduction sets the stage by highlighting the significance of post-operative pain management and the role of peripheral nerve blockade, particularly lumbar plexus block (LPB), in addressing this challenge. To enhance this section, you could provide more context on the prevalence and impact of post-operative pain in hip and femur orthopedic surgeries. Additionally, discussing the limitations and challenges associated with current pain management approaches could provide a stronger rationale for the study.

Methods:

The methods section provides a detailed description of the study's design, search strategy, eligibility criteria, study selection process, quality assessment, data extraction, and analysis methods. To improve clarity, consider organizing the subsections more logically and providing more details on specific aspects, such as the criteria used for risk of bias assessment and data extraction. Additionally, including a flowchart summarizing the study selection process could enhance transparency.

Results:

The results section presents the findings of the literature search, study selection process, and key outcomes of the included studies. To enhance readability, consider organizing the results into subsections corresponding to the main themes or outcomes assessed in the study (e.g., pain scores, opioid use, adverse effects). Additionally, providing more context or interpretation for the presented data could help readers better understand the significance of the findings.

Discussion:

The discussion section interprets the study findings in the context of existing literature and provides insights into the implications of the results. To strengthen this section, consider discussing the clinical relevance of the findings in more detail, including how they contribute to current understanding and practice in post-operative pain management. Additionally, addressing potential limitations and future directions for research could provide a more comprehensive analysis.

Conclusion:

The conclusion summarizes the key findings of the study and emphasizes its implications for clinical practice and future research. To enhance this section, consider reiterating the main advantages of continuous lumbar plexus blockade identified in the study and emphasizing their potential impact on patient outcomes and healthcare practices. Additionally, highlighting any remaining gaps in knowledge or areas for further investigation could provide a more balanced conclusion.

Author Response

Thank you for your feedback. Detailed point by point response is uploaded as a pdf file. 

Reviewer 2 Report

Comments and Suggestions for Authors

Review

Many thanks to the authors for having presented a so interesting systematic review about “Efficacy of Continuous Lumbar Plexus Blockade in Managing Post-operative Pain After Hip or Femur Orthopedic Surgeries: A Systematic Review and Meta-Analysis”.

Before resubmitting the revision version of the article, please read the editorial rules carefully, and check other editorial aspects, such as: text alignment (lacking), text justification at the head (lacking), etc. The language is so good that the manuscript does not need to be corrected by a person of English mother tongue.

Plagiarism

Detected plagiarism:  17% (it should be < 15%). The authors must be reduce this percentage.

Title and Abstract

The title and abstract cover the main aspect of the work and capture the appropriate essence of the manuscript. Theabstract is quite well structured, and it contains the main results of the study.

Key words

They are chosen correctly.

Background

The introduction is quite well structured, containing the information relevant to the study and the main aims of the study. Please, briefly define the pros and cons of traditional opioid-based regimens.

Please specify what types of anesthetics were used to perform the various procedures.

Lines 54-55: “Post-operative pain is common in hip and femur orthopedic surgeries, impacting patient recovery and potentially causing complications”.

Please, add few lines regarding how these aspects impact elderly patient recovery and quote:

·      https://pubmed.ncbi.nlm.nih.gov/31698279/

Lines 62-64: “The study results indicated that LPB combined with light sedation can be considered an effective, safe, and acceptable anesthetic option for managing hip fractures in older people by preserving hemodynamic stability.”Please, delineate what you mean by “light”.

Methods

This section contains enough information to understand and possibly repeat the study. 

Inclusion and exclusion criteria are clear. The experimental design is valid and there are an adequate number of studies and patients to justify the findings.

Statistical analysis 

The statistical analysis is appropriate to the research.

Results

The results presented are quite complete, reflecting the MM section. 

The results are reproducible, reflect clinical expectations and are displayed legibly.

Discussion

The length and content of the discussion communicate the main information of the paper. 

The review provides valuable input to the field.

I think that a summary table with various pros and cons of each method can help to acquire a better overall vision.

Conclusions

The conclusion provides a clear summary of the main points and presents their meaning.

The conclusions reflect and refer to the results of the study and is justified by the results and methods.

References

References are relevant to the study and in the correct style.

The references are up to date and relevant, but they should be integrated as suggested previously. 

Please add the doi and the PMID in the citations to facilitate the access of the reviews.

Competing interest

The interests of the authors did not create a bias in the reporting of results and conclusions.

Concerns

The manuscript or study raised no ethical concerns.

Tables and Figures

The number and quality of tables are appropriate to transmit the main information of the paper. 

Comments on the Quality of English Language

Minor editing of English language required.

Author Response

Dear reviewer,

Thank you for your comments and valuable feedback. 

Edits have been made as per your comments to the introduction and highlighted in yellow in the revised manuscript. Also, plagiarism has been reduced to < 15%.

Best regards, 

Wijdan AlMutiri 

Reviewer 3 Report

Comments and Suggestions for Authors

The manuscript reflects commendable work and is well-written. However, there are minor concerns that need to be addressed to ensure compliance with academic and publishing standards.

Specific comments:

Page 3, Lines 84-86: It is advised to include the complete search algorithm utilized for each database searched.

Page 3, Lines 110-115: Please provide a more extensive explanation of the statistical analysis conducted.

Page 4, Line 122: It is recommended to cite the papers considered for inclusion and elucidate reasons for their exclusion.

Page 11, Lines 304-312: A comprehensive section detailing the limitations of the study is absent. This aspect warrants attention.

Author Response

Dear reviewer, 

We would like to sincerely thank you for your advice and constructive comments that allowed us to greatly improve the quality of the manuscript. Added/modified sentences have been yellow-highlighted in the attached corrected manuscript. Please find our point-by-point detailed response below

  1. Done in the Search and Identification of Studies subsection of the methods.
  2. Done in the data analysis subsection of the methods.
  3. Attached is a supplementary file containing the study title, first author, year of publication and reason for exclusion. Also, as we revised the reasons for exclusions, we found more proper reasons, so we adjusted the flow diagram.
  4. We appreciate the reviewer's observation and valuable feedback regarding the absence of a comprehensive section detailing the limitations of the study in this section. We have made the necessary revisions to the manuscript by moving the discussion of the study's limitations to this dedicated section.

best regards,

Wijdan AlMutiri 

Round 2

Reviewer 1 Report

Comments and Suggestions for Authors

I am satisfied with the authors responses to my queries

Author Response

Thank you for your replay.

Reviewer 2 Report

Comments and Suggestions for Authors

This comment was not addressed by the authors. Please, provide it improving references section as well. 

Background

The introduction is quite well structured, containing the information relevant to the study and the main aims of the study. Please, briefly define the pros and cons of traditional opioid-based regimens.

Please specify what types of anesthetics were used to perform the various procedures.

Lines 54-55: “Post-operative pain is common in hip and femur orthopedic surgeries, impacting patient recovery and potentially causing complications”.

Please, add few lines regarding how these aspects impact elderly patient recovery and quote:

·      https://pubmed.ncbi.nlm.nih.gov/31698279/

Comments on the Quality of English Language

Minor editing of English language required

Author Response

Good day,

See the attached point-by-point response file

Thank you for your review and valuable comments

Best regards,

Wijdan AlMutiri 
